# Effect of habitat fragmentation on rural house invasion by sylvatic triatomines: A multiple landscape-scale approach

Miriam Cardozo[1]*, Federico Gastón Fiad[1], Liliana Beatríz Crocco[1], David Eladio Gorla[2]

1 Universidad Nacional de Córdoba, Facultad de Ciencias Exactas, Físicas y Naturales, Cátedra de Introducción a la Biología, Instituto de Investigaciones Biológicas y Tecnológicas (IIBYT-CONICET), Córdoba, Argentina, 2 Universidad Nacional de Córdoba, Grupo de Ecología y Control de Vectores, Instituto de Diversidad y Ecología Animal (IDEA- CONICET), Córdoba, Argentina

☉ These authors contributed equally to this work.
* cardozo.miri@gmail.com

**Data Availability Statement:** All relevant data are within the manuscript and its Supporting Information files.

## Abstract

After the decrease of the relative importance of *Triatoma infestans*, a number of studies reported the occurrence of sylvatic triatomines dispersing actively to domestic environments in the dry western Chaco Region of Argentina. Anthropic modification of the landscape is mentioned as one of the main causes of the increase in domicile invasion. The aim of this study was to describe the occurrence and frequency of sylvatic triatomines invading rural houses, and to evaluate the effect of habitat fragmentation and other ecological factors on the invasion of rural houses in central Argentina. We hypothesized that the decrease in food sources and the loss of wild ecotopes, as a consequence of habitat fragmentation, increase the chances of invasion by triatomines. The entomological data was collected by community-based vector surveillance during fieldwork carried out between 2017–2020, over 131 houses located in fourteen rural communities in the northwest of Córdoba Province (central Argentina). We used generalized linear models to evaluate the effect of (i) the environmental anthropic disturbance in the study area, (ii) the composition and configuration of the landscape surrounding the house, (iii) the spatial arrangement of houses, (iv) and the availability of artificial refuges and domestic animals in the peridomicile, on house invasion by triatomines. We report the occurrence of seven species of triatomines invading rural houses in the study area -*T. infestans*, *T. guasayana*, *T. garciabesi*, *T. platensis*, *T. delpontei*, *T. breyeri* and *P. guentheri*-. Study data suggest that invasion by triatomines occurs with higher frequency in disturbed landscapes, with houses spatially isolated and in proximity to subdivided fragments of forest. The availability of domestic refuges in the peridomestic structures as well as the presence of a higher number of domestic animals increase the chances of invasion by triatomines.

**Funding:** This study received financial support from The National Agency for Scientific and Technological Promotion (ANPCyT), through the Fund for Scientific and Technological Research (FONCyT), PICT 2016-2527 (DG). URL: https://www.argentina.gob.ar/ciencia/agencia/fondo-para-la-investigacion-cientifica-y-tecnologica-foncyt The funders had no role in study design, data collection and analysis, decision to publish, or preparation of the manuscript.

**Competing interests:** The authors have declared that no competing interests exist.

## Author summary

*Triatoma infestans* is the main vector of *Trypanosoma cruzi*, the causal agent of Chagas disease in the southern cone countries of South America. A number of sylvatic species of triatomines have been reported dispersing actively into man-made habitats. The study of the ecological mechanisms associated with the active dispersion of sylvatic triatomines into domestic environments becomes relevant in terms of the potential risk of *T. cruzi* introduction from the wild cycle into the domestic transmission cycle. Anthropic modification of the landscape is mentioned as one of the possible causes of the increase in the active dispersion, due to the negative impact of habitat fragmentation on the availability of food sources (populations of birds and mammals) and wild ecotopes for sylvatic triatomines. Before the present study there have been no evidences testing this hypothesis. Previous authors focused on determining the effect of environmental disturbance considering one geographical scale. In this article we describe the occurrence and frequency of sylvatic triatomines invasion into rural houses, and discuss the effect of habitat fragmentation considering a multiple landscape-scale approach. This approach allowed testing the effect of compositional and configurational landscape properties around the house and environmental anthropic disturbance, among other ecological factors.

## Introduction

Chagas disease is caused by the parasite *Trypanosoma cruzi*, which also infects more than 100 species of domestic and sylvatic mammals and can be transmitted by blood-sucking insects of the subfamily Triatominae (Reduviidae) [1].

Most of the 140 triatomine species have been shown to be naturally or experimentally infected with *T. cruzi*. However, relatively few have epidemiological importance [2].

In the last years, several studies reported the occurrence of sylvatic triatomines dispersing actively to domestic environments in Argentina. Various species were reported invading houses of the dry western Chaco Region: *Triatoma infestans* (the main domestic vector of Chagas disease in Latin America), *Triatoma garciabesi*, *Triatoma guasayana*, *Triatoma platensis*, *Triatoma patagonica*, *Triatoma eratyrusiformis*, *Triatoma sordida* and *Panstrongylus guentheri* [3–7].

Although sylvatic species are considered of secondary epidemiological importance, they are responsible for maintaining the sylvatic transmission cycle of *T. cruzi*, involving several species of wild mammals, including marsupials, carnivores, armadillos, bats and rodents [8–10]. Previous studies carried out in the semiarid Chaco region of Argentina reported high infection prevalence in opossums and armadillos [11,12] and high infection rates with *T. cruzi* from sylvatic triatomines invading rural houses [13]. Therefore, understanding the underlying mechanisms of domicile invasion by sylvatic triatomines becomes relevant in terms of the introduction of *T. cruzi* from the sylvatic to domestic transmission cycle and the associated risk of Chagas disease transmission to humans [9,14].

Previous studies provided evidence that sylvatic triatomines invasion in the domestic and peridomestic environment are associated with the type of construction and materials used in outbuildings for storage or housing animals (e.g., chicken coops and goat corrals), the presence of mammalian or avian hosts in the peridomicile area, housing lights and the proximity to sylvatic areas [15–18].

Anthropic modification of the landscape (particularly habitat fragmentation) has also been suggested as one of the main causes of sylvatic triatomines dispersal to domestic habitats [19].

Habitat fragmentation is defined as a landscape-scale process that involves both habitat loss and habitat breaking apart into smaller and more isolated patches [20]. Changes in the composition and spatial configuration of habitat patches, due to fragmentation of the native vegetation, drastically affect the stability of local flora and fauna, causing disturbances in the ecosystem [21,22]. The association between wild populations of triatomines and habitat fragmentation is indirect. It is related mainly to the effect of environmental disturbance on the food source decrease for sylvatic triatomines (birds and mammal's populations), as well as a decrease of wild refuge availability (e.g., nests, burrows, tree bark, dry cacti, fallen logs). The effect of fragmentation on the population dynamics of vertebrates can be evaluated according to three fundamental components: habitat patch size, degree of isolation between patches and quality of the habitat in that fragment [23,24]. Studies evaluating bird and mammal populations in fragmented environments observed that decrease in patch area is associated with decreased food sources and with increased parasitism rates or predation, while the degree of isolation and subdivision of the patches can lead to less success of dispersal between patches [24–26]. The decrease of local populations of birds and mammals as well as the damage and loss of wild ecotopes, as a consequence of environmental disturbance, results in adverse conditions for sylvatic populations of triatomines. This critical condition activates survival strategies in insects related to habitat choice, reproductive behavior and search for blood sources, triggering the active dispersal and invasion of human dwellings [27,28].

The invasion of human dwellings by sylvatic triatomines has been little studied, particularly in the southern extreme of the Argentinean Gran Chaco Region. As part of a broader study conducted in the northwest of Córdoba province (central Argentina), we describe the invasion occurrence and frequency of sylvatic triatomines into rural houses, and we evaluate the factors driving the invasion process. We hypothesized that the active dispersal of triatomines is strongly related to the search for food and the selection of refuge [29], so alterations in the availability of food sources and refuge, due to the effect of the landscape fragmentation, increase the chances of domicile invasion.

Using a multi-model inference approach and entomological 4 year-field data, we found that environmental anthropic disturbance, measured at different landscape scales, artificial refuge availability, number of domestic animals, as well as the spatial aggregation of houses, have potential effects on the invasion of sylvatic triatomines.

## Materials and methods

### Ethics statement

All adult inhabitants of houses involved in the study received detailed explanations about the project and instructions on how to collect all insects resembling triatomines.

All adults invited accepted to participate, giving their consent in oral form. Oral consent of each member of the community was recorded in the data sheets. There is no institutional review board (IRB) in our institutions, neither of CONICET nor the University of Córdoba.

### Study area

The study area is located in the southern extreme of the Gran Chaco region, between latitudes 30˚ and 31˚ S and longitudes 64˚ and 65˚ W. This region belongs to the phytogeographic province of Chaco and it is characterized by a dry subtropical climate, with hot summers (>40˚C) and cold winters (<-5˚C) [30,31]. The study was carried out between 2017 and 2020, over 131 houses located in fourteen rural communities of Cruz del Eje and Ischilín Departments (NW Córdoba province, central Argentina) (Fig 1). Most of the traditional rural houses in the study area (historically built with adobe walls and local vegetation on the roofs) were replaced, in the

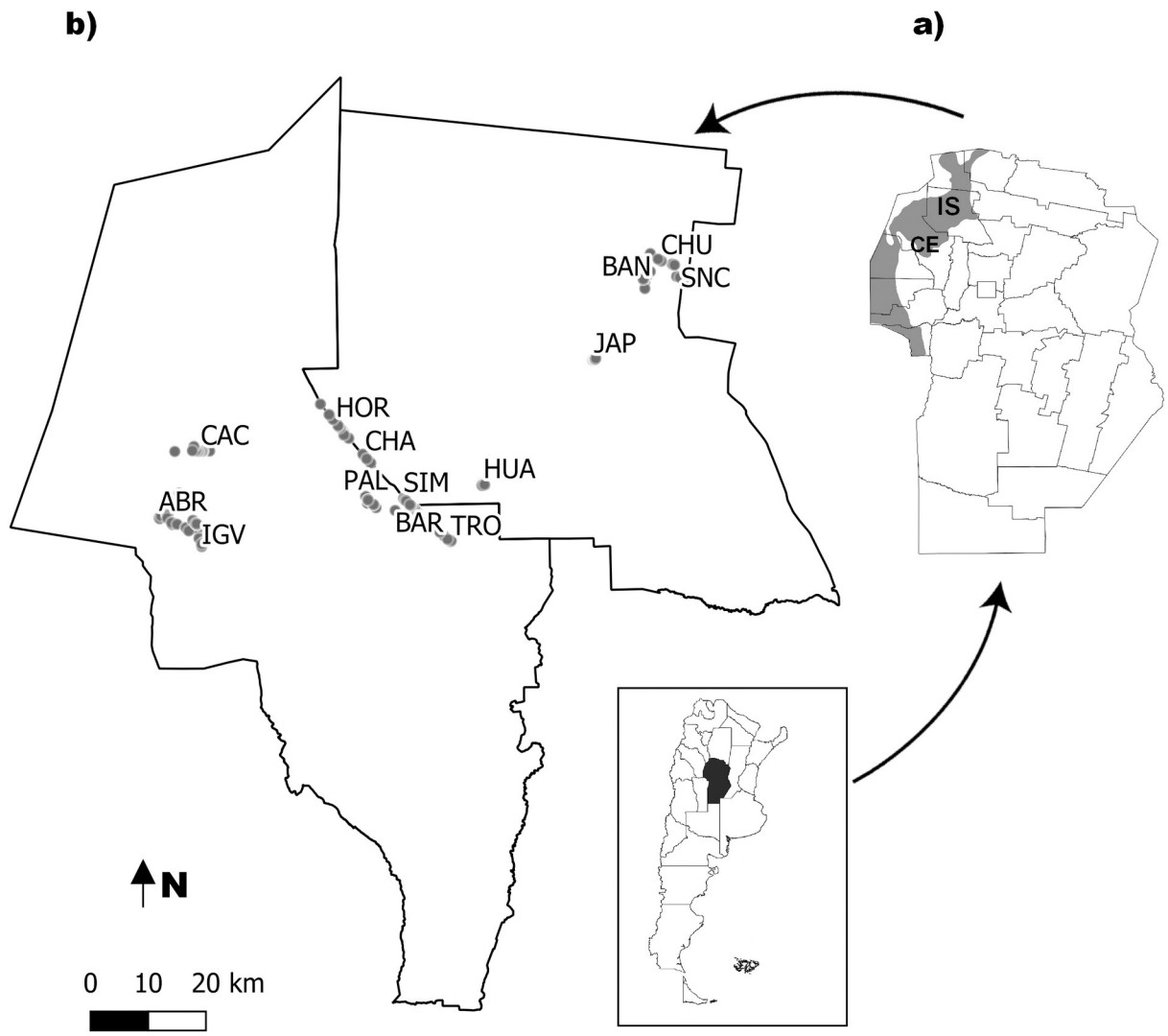

**Fig 1. Study area. A)** The map shows the departmental boundaries of Córdoba (Argentina), the location of the Departments Cruz del Eje (**CE**) and Ischilín (**IS**) and the Chaco phytogeographic region shaded in gray. The base map layer was provided by Instituto Geográfico Nacional (https://www.ign.gob.ar) **B)** Location of the rural houses (grey points) and localities sampled during 2017–2020. ABR Las Abras, IGV Iglesia Vieja, CAC Cachiyuyo, HOR Hormigueros, CHA Chañaritos, SIM El Simbolar, TRO El Tropiezo and PAL Palo Parado, BAR Barrial, HUA Huascha, JAP Jaime Peter, SNC San Nicolás de las Chacras, BAN El Bañado and CHU Los Churquis.

last decade, by dwellings with brick plastered walls and galvanized aluminum or cement roof. Although all rural houses visited during field work had electricity, the use of mosquito nets on doors and windows is very rare.

The vegetation composition in the study area is characterized by a mixture of open and closed xerophytic forests and shrublands, 2–4 m in height [32]. For several decades, the semi-arid Chaco region of Argentina has been undergoing intense environmental changes, mainly due to the expansion of agriculture. Between 1970 and 2000, more than 1,000,000 ha of dry forests were lost in Córdoba province, with deforestation rates similar or even higher than those recorded for tropical forests. Approximately 85% of the originally undisturbed Chaco forest of the northern of Córdoba province has been transformed into a highly fragmented mosaic of isolated patches of forest, dense thorny shrubs, seminatural grassland and a large

surface of cultivated land. These modifications varied in intensity, with an east-west gradient associated with the rainfall regime [33].

## Entomological data

The frequency and occurrence of sylvatic triatomines invading rural houses were recorded through community-based vector surveillance, in which the owners recorded the arrival of triatomines into their houses. The inhabitants who agreed to participate in the study and gave their oral consent, were trained in the identification and careful collection of triatomines to avoid accidental infection by *T. cruzi* in the manipulation of the triatomines. During fieldwork, each family received plastic bags to collect dispersing triatomines arriving to the house attracted by the light and the ones they found inside or in the surroundings of the house. After 15 days a new visit was made to collect the bags and record the catches. Triatomine collection was carried out five times, at the beginning of the warm season (November and December 2017, 2018 and 2019), and at the end of the warm season (February and March 2019 and 2020). Field sampling was conducted during the warm season (November—March), because active dispersion of triatomines in the Chaco region is more frequent during the warmer months of the year. The sensitivity of the community-based surveillance method was corroborated in previous studies [34,35].

The collected triatomines were taxonomically identified [36] in laboratory. For the present analysis, the data on occurrence (presence/absence) of sylvatic triatomines collected in rural houses were divided into two groups. In one group we included the occurrence of sylvatic triatomines associated mostly with birds: *T. garciabesi*, *T. platensis* and *T. delpontei*. These species inhabit wild ecotopes associated with birds, such as nests of Furnariidae and Psittacidae, and in the peridomestic environment they can be found in chicken coops [36–39]. In the second group, we included the occurrence of triatomines associated mostly with mammals: *T. guasayana*, *P. guentheri* and *T. breyeri*. These species inhabit ecotopes associated with rodents (Caviidae and Cricetidae), marsupials (Didelphidae), in dry cacti of the genus *Opuntia* spp. and in goat or sheep pens [36,39–41]. The data on the frequency of invasion of triatomines associated with birds and mammals were used as proportions with a binomial error distribution, considering the number of times the house recorded the presence of triatomines, over the total number of times that the house was visited during the fieldwork.

## Land cover data

To characterize the land cover of the study area, we used a Landsat 8 OLI satellite image from January 2018 (Path / Row 230–81, 30 x 30 m resolution), provided by the US Geological Survey (USGS) (https://earthexplorer.usgs.gov/). First, the image was radiometrically calibrated and mountainous and saline areas were eliminated from the scene. Then, the image was analyzed through an unsupervised cluster technique (ISODATA) [42]. Five types of land cover were defined, considering the unsupervised classification and previous studies on the area [32,43]; closed forest, open forest, open shrubland, closed shrubland, and cultural land (roads, bare soil, cities, and croplands). The composition of the landscape was characterized by supervised classification using the maximum likelihood method [42]. The classification methods used to obtain the thematic map were selected considering the methodology used in the latest update of floristic composition of native woody vegetation types in the southern Great Chaco and Espinal [32]. Validation of the thematic map was performed using field data and visual interpretation of historical photos available in Google Earth. The confusion matrix method was used to estimate the accuracy of the classification. The software ENVI 5.1 was used for the preprocessing and processing of the image [44].

## Characterization of environmental anthropic disturbance

In order to compare the habitat fragmentation in the study area, we defined areas with three different degrees of environmental anthropic disturbance: preserved, intermediate and disturbed landscape (Table 1).

To this end, first the coverage classes of the thematic map were grouped into two categories: forest-shrubland cover and anthropized cover. The first one comprises closed forest, open forest and closed shrubland covers; and the second one includes open shrubland and cultural land.

Second, the localities were grouped into eight circular zones of 7 km radius and for each zone three compositional and configuration landscape metrics were extracted in order to evaluate the degree of environmental intervention (Fig 2A). We measured (i) the total coverage area (Ha.) of both classes, (ii) the connectance index (proportion of functional joining between patches of forest-shrubland according to a critical distance of 200 m) and (iii) the landscape DIVISION index for the forest-shrubland cover (probability that two pixels chosen randomly in the landscape do not belong to the same patch). The extraction of the metrics was carried out using the software Fragstat 4 [45].

## Characterization of the landscape environment

To characterize the landscape environment at a micro scale, we considered 4 circular areas around each house, of 500, 1000, 2000 and 5000 meters in diameter (Fig 2B). For each area, landscape metrics were extracted, in order to evaluate the three fundamental components of

**Table 1. Categories of analyzed effects and their predictors used in the modelling.**

| Effect | Predictor | Description |
|---|---|---|
| Environmental anthropic disturbance | Landscape disturbance | Landscape classification according to the anthropic environmental disturbance; *Preserved landscape*: >60% of forest-shrubland cover, high connectance index between forest patches (>0.67) and low DIVISION index (<0.33); *Intermediate landscape*: 40–50% of forest-shrubland cover and anthropized cover, intermediate values of connectance index and forest DIVISION index (between 0.34–0.66) and *Disturbed landscape*: >60% of anthropized cover, low connectance index between forest patches (<0.33) and high DIVISION index (>0.67). |
| Landscape environment around the house (*) | Forest-shrubland cover | Area (Ha.) covered by original forest-shrubland in good condition. |
| | Mean patch area | Mean patch area (Ha.) of forest-shrubland cover. |
| | Connectance Index | Proportion of functional joining between patches of forest-shrubland, where each pair of patches is connected or not according to a distance criterion of 200 meters. Values range from 0 to 1, a value of 0 means that no patches are attached and values of 1 means all patches are attached. |
| | DIVISION Index | Probability that two pixels chosen randomly in the landscape do not belong to the same patch. Values range from 0 to 1, the higher the value the more subdivided the landscape is (more habitat fragmentation). |
| | Patch density | Density of patches of forest-shrubland cover (no / 100 Ha.). |
| Spatial house arrangement | Density | Density of houses within a radial distance of 450 meters (no. houses / Ha.). |
| | Distance | Minimum distance from each house to the nearest house (meters). |
| Availability of artificial refuges and domestic food sources | No. artificial refuges | Number of peridomestic structures recorded in the house: corrals for goats, pigs, horses or cows, chicken coops, logs where chickens sleep or other domestic birds, hutches, firewood piles and storage rooms. |
| | No. chickens | Number of chickens reported in the house. |
| | No. goats | Number of goats reported in the house. |
| | No. dogs | Number of dogs reported in the house. |
| | No. humans | Number of inhabitants reported in the house. |

(*) These landscape metrics were measured in four circular areas of 500, 1000, 2000 and 5000 meters in diameter around the house.

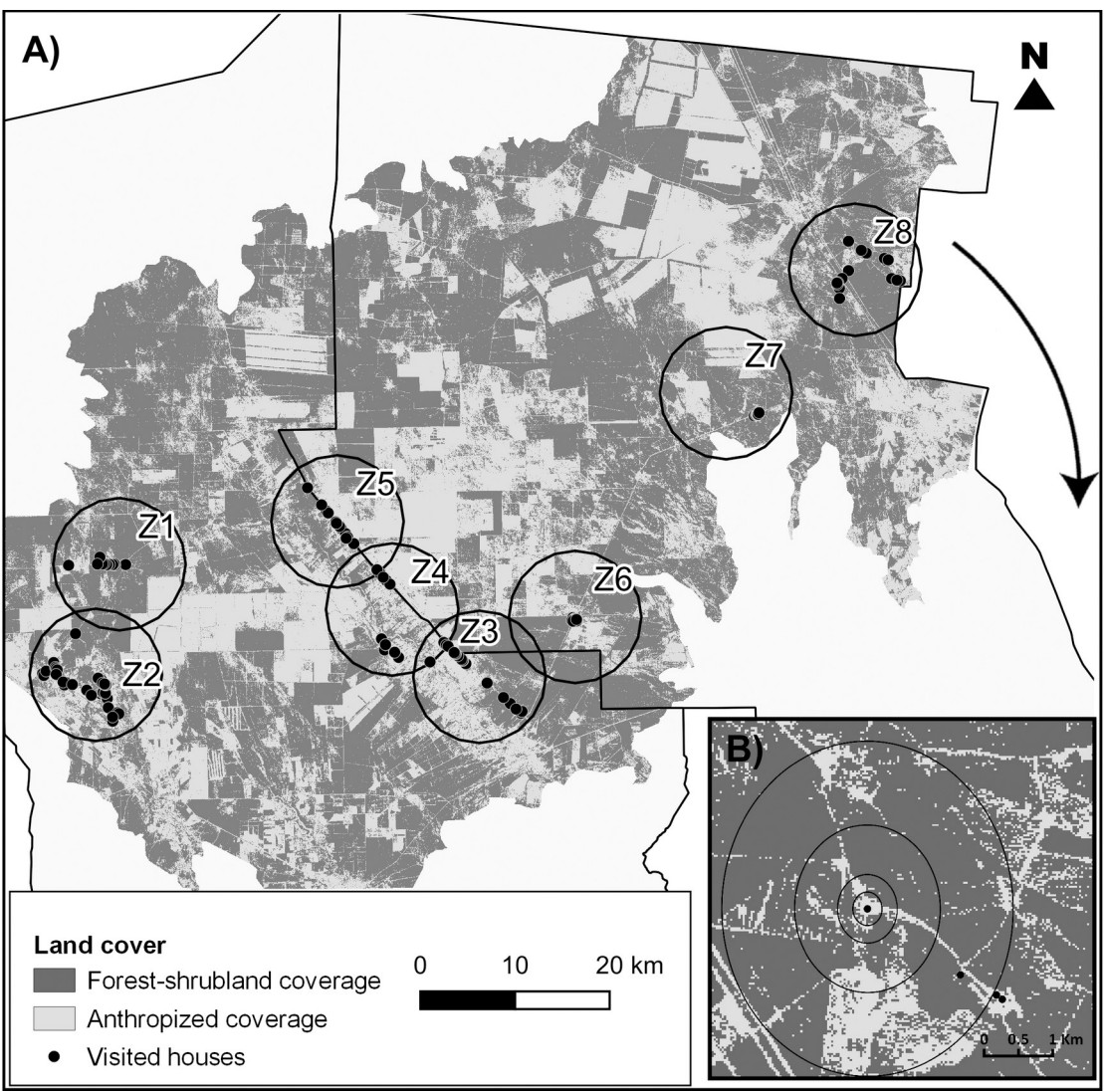

**Fig 2. A) Characterization of environmental anthropic disturbance.** The map shows the land covers obtained through the supervised classification of the LANDSAT satellite image (http://www.usgs.gov). The black points correspond to the surveyed houses and the circles delimit the zones from which the landscape was characterized in the study area. **B) Characterization of landscape environment**. The concentric circles correspond to the areas of influence of 500, 1000, 2000 and 5000 meters in diameter around each house from which the landscape metrics were extracted.

fragmentation [23]: (i) area of the fragment, measured through mean patch size (Ha) and density of forest-shrubland patches (no. of patches / 100 Ha), (ii) degree of isolation, evaluated through the connectance index and the DIVISION index, and (iii) quality of habitat in that fragment, evaluated through the forest-shrubland cover (Ha) remaining in the area (Table 1).

## Spatial arrangement of houses

In order to evaluate the possible effect of the spatial arrangement of houses in the area, we calculated the density of houses (number of houses / Ha.) in a radius of 450 m around each sampled house and the minimum distance (meters) to the closest house. Both variables were calculated using the QGIS 3.4.6 software (http://qgis.osgeo.org) [46].

## Availability of artificial refuge and domestic food sources

We assessed the effect of artificial refuge availability for sylvatic triatomines offered by the peri-domestic structures and the number of domestic hosts present in the peridomicile and domicile, since they represent potential food sources for the triatomines. For this purpose, we recorded the number of peridomicile structures (such as corrals for goats, pigs, horses or cows, chicken coops, logs where chickens sleep or other domestic birds, hutches, firewood piles and storage room) and the number of goats, chickens, dogs and inhabitants in the house, through interviews with the local families (Table 1).

## Data analysis

Exploratory analysis of the data was performed in order to observe the relationships between variables, and evaluate the correlation using Spearman correlation. Prior to the analysis, the explanatory variables were standardized with mean of zero and a standard deviation equal to one.

The possible spatial dependence of the data was evaluated using a spline correlogram [47] with a 95% confidence interval.

Based on the accumulated knowledge about the ecology of triatomines and the effect of fragmentation on the populations of birds and mammals, we postulated *a priori* hypothesis to evaluate the integrated effect of explanatory variables, measured at multiple scales, on the occurrence and frequency of invasion by sylvatic triatomines. For the analysis of the response variables, generalized linear models (GLM) with binomial error distribution and logit link function were used. We built 65 and 69 models to evaluate the occurrence of triatomines associated with birds and mammals, respectively, and 55 models to evaluate the invasion frequency of bird-associated species and 58 models for the invasion frequency of mammalian-associated species. The adjusted models for each response variable were ordered into 6 categories:

1. Null Model, representing the hypothesis of the randomness of the occurrence and frequency of invasion in the study area.

2. Landscape disturbance models, representing the hypothesis that habitat fragmentation decreases the environmental supply of food sources and refuge for sylvatic triatomines [14,48,49] causing their active dispersal [29,50]. Were the statement true, we expected a higher occurrence and frequency of invasion in landscape of intermediate disturbance (where the environmental instability drives the dispersion of triatomines), followed by preserved environments (where triatomines populations may be closer to equilibrium) and finally, disturbed environments (where triatomine populations are in suboptimal conditions resulting in a lower number of individuals dispersing).

3. Focal Models, representing the hypothesis that the spatial configuration of forest-shrubland fragments surrounding the house is related with the chances of finding local populations of vertebrates and wild refuges that are suitable for the establishment of triatomines. Therefore, it is expected to find higher triatomine invasion in houses surrounded by landscapes with (i) higher average size of forest patches (ii) higher connectivity between forest patches, (iii) less subdivision of the landscape, (iv) intermediate density values, of forest patches and (v) intermediate cover values of forest-shrubland.

4. Models of spatial configuration of houses, representing the hypothesis that triatomines invasion is higher when houses are spatially clustered, because of the spatial concentration of physical and chemical signals (e.g. public lights, domestic hosts odors).

5. Models of refuge availability and domestic food sources, representing the hypothesis that active dispersal is mediated by the search for food sources and selection of refuge, so

triatomines invasion should be more frequent in houses with larger number of domestic hosts and peridomestic structures.

6. Combined models, containing different combinations of the predictors guided by *a priori* hypotheses.

To avoid multicollinearity between variables, the variance inflation factor (VIF) was calculated for each model, discarding the models with VIF value > 10 [47].

A multi-model inference approach based on the Akaike information criterion (AICc) was used to estimate the effect of the predictors and their relative importance on the response variables [51]. First, models were selected according to AICc -lower values indicate better compromise between model fit and model complexity-. Second, coefficient estimates for each variable and its 95% confidence interval were estimated averaging the coefficient value in each model where the variable was present. Subsequently, the regression between the estimated coefficients of the predictors with major effects on the response variables were plotted using the *visreg* package in the R software [52].

Model validation for the invasion occurrence was performed using the K-fold cross validation method. This approach involves randomly dividing the set of observations into k groups, of approximately the same size. The first group is treated as a validation set and the remaining k-1 groups are used to train the model. This procedure is repeated k times, obtaining as a result the average estimate of the prediction error [53]. To estimate the prediction error of the selected models, we used a K value of 10 and we considered a classification threshold probability of 0.5.

Model validation for the invasion frequency models was performed using the area under the receiver operating characteristic curve method (AUC-ROC) for multiple classes, using the *multiclass.roc* function of the pROC package. The AUC values vary between 0 and 1, a model that correctly separates the classes would have 100% sensitivity and specificity, yielding an AUC value of 1, while a model as good as pure chance would result in AUC values close to 0.5 [53].

Finally, the performance of the invasion occurrence and frequency models was evaluated using a confusion matrix, from which we calculated overall model accuracy rate (percentage of correctly classified observations out of the total), sensitivity (proportion of true positives out of total positive predictions) and specificity (proportion of true negatives out of total negative predictions) [54]. We conducted the analyses of spatial autocorrelation, model selection and validation using the R software version 3.4.3 [55].

## Results

### Descriptive results

There was a moderate level of correlation between the explanatory variables. The forest cover metrics measured at different scales (500, 1000, 2000 and 5000 m) showed the highest correlation (r >0.70). To reduce the correlation, we recast the forest cover metrics as a linear combination between them. Hence, the area of 2000, 1000 and 500 meters in diameter was subtracted from the area of 5000, 2000 and 1000 meters, respectively, so that in the analysis we considered vegetation cover rings around the house. The variables of anthropized cover, measured at different scales, were not considered in the model building. No spatial correlation was observed between the data, according to the spline correlograms obtained for each response variable, (r <0.5).

Table 2 summarizes the data corresponding to the localities evaluated during fieldwork conducted between 2017–2020. It also presents the grouping of the localities in the 8 circular

**Table 2. Descriptive features of the localities evaluated during fieldwork conducted between 2017–2020 in northwest Departments of Córdoba Province.**

| Landscape disturbance | Locality | Zone | No. houses evaluated | No. of peridomicile structures | Mean number per house | | | |
|---|---|---|---|---|---|---|---|---|
| | | | | | goats | chicken | dogs | humans |
| Preserved landscape | Cachiyuyo | Z1 | 12 | 26 | 51.4 | 23.3 | 2.3 | 3 |
| | Jaime Peter | Z7 | 10 | 15 | 2 | 6.1 | 1.6 | 3 |
| | San Nicolás de las Chacras | Z8 | 7 | 13 | 52.8 | 17.1 | 4.4 | 3 |
| | El Bañado | Z8 | 8 | 13 | 13 | 10 | 3.5 | 4.25 |
| | Los Churquis | Z8 | 5 | 14 | 38 | 9.6 | 3 | 3.2 |
| | **Total** | - | **42** | **81** | **1302** | **596** | **118** | **136** |
| Intermediate landscape | El Tropiezo | Z3 | 13 | 23 | 7.7 | 6.4 | 4.6 | 3 |
| | El Barreal | Z3 | 1 | 0 | 0 | 0 | 4 | 12 |
| | Huascha | Z6 | 7 | 8 | 14.4 | 5.7 | 2 | 3 |
| | **Total** | - | **21** | **31** | **202** | **119** | **75** | **70** |
| Disturbed landscape | Las Abras | Z2 | 8 | 25 | 2.3 | 21 | 2.8 | 3.5 |
| | Iglesia Vieja | Z2 | 25 | 34 | 0.8 | 7.1 | 2.8 | 3.2 |
| | Chañaritos | Z4/Z5 | 14 | 11 | 0 | 7.8 | 1.9 | 3.4 |
| | Hormigueros | Z5 | 8 | 24 | 23.5 | 13.7 | 2.6 | 2.7 |
| | El Simbolar | Z4 | 5 | 9 | 0 | 9.2 | 3.4 | 4.2 |
| | Palo Parado | Z4 | 8 | 4 | 0.6 | 1.8 | 1.5 | 4.5 |
| | **Total** | - | **68** | **107** | **233** | **607** | **166** | **231** |

zones and the degree of environmental disturbance in which they were classified into, as well as the number of peridomestic structures and the mean number of domestic animals recorded per house.

## Entomological data

Through community-based surveillance, 678 adult triatomines of seven species were found invading rural houses in the study area; *T. infestans*, *T. guasayana*, *T. garciabesi*, *T. delpontei*, *T. platensis*, *T. breyeri* and *P. guentheri* (Table 3).

In general, a higher fraction of positive houses reported the invasion of mammalian-associated species (60.3%; 79/131) during field surveys compared with bird-associated species (40.5%; 53/131). The co-occurrence of both groups of species was observed in 40.5% (53/131) of the domiciles and the presence of *T. infestans* was recorded in 30.5% (40/131) of houses. Raw data are available in S1 Data.

**Table 3. List of triatomines captured by community-based surveillance in northwest of Córdoba Province during 2017–2020.**

| Group | Species | No. captured |
|---|---|---|
| Bird-associated species | *Triatoma garciabesi* | 167 |
| | *Triatoma platensis* | 18 |
| | *Triatoma delpontei* | 5 |
| Mammalian-associated species | *Triatoma guasayana* | 367 |
| | *Panstrongylus guentheri* | 23 |
| | *Triatoma breyeri* | 3 |
| Other | *Triatoma infestans* [a] | 95 |

[a] Of the 95 *T. infestans* specimens captured, 21 correspond to nymphs. These captures were not analyzed in the present study

**Table 4. Best- ranked models structure according to ΔAICc for each response variable.**

| Response variable | Model | ΔAICc | Model predictors |
|---|---|---|---|
| Invasion occurrence of bird-associated species | m41 | 0.00 | no. refuges + no. chickens |
| | m57 | 0.19 | landscape disturbance + no. refuges + no. chickens |
| | m38 | 2.04 | no. refuges |
| | m49 | 2.97 | landscape disturbance + forest cover 500m + mean patch area 500m + subdivision 500m + connectivity 500m + house density + distance |
| Invasion occurrence of mammalian-associated species | m49 | 0.00 | landscape disturbance + forest cover 1000m + mean patch area 1000m + subdivision 1000m |
| | m62 | 2.02 | landscape disturbance + no. refuges + no. humans |
| | m4 | 2.05 | landscape disturbance |
| | m30 | 2.19 | forest cover 1000m + mean patch area 1000m + subdivision 1000m |
| | m6 | 2.35 | forest cover 1000m |
| | m46 | 2.44 | landscape disturbance + house density |
| | m14 | 2.75 | mean patch area 1000m |
| | m43 | 2.81 | no. refuges + no. humans |
| Invasion frequency of bird-associated species | m39 | 0.00 | landscape disturbance + forest cover 500m + mean patch area 500m + subdivision 500m + connectivity 500m + house density |
| Invasion frequency of mammalian-associated species | m58 | 0.00 | house density + no. goats + no. humans |
| | m34 | 1.47 | no. refuges + no. goats |
| | m30 | 2.02 | no. goats |

AICc, second order Akaike information criterion.

ΔAICc, is the difference with the best ranked model.

According to the owners, most of the individuals were captured around the lights from the exterior walls of the houses, although many also reported capturing them inside the house, behind wall paintings or under the bed.

## Invasion occurrence: Bird-associated species

A total of 40.5% (53/131) of the evaluated houses showed presence of bird-associated triatomines invading the domestic environment. The species most frequently recorded invading houses were *T. garciabesi* (167), followed by *T. platensis* (18) and *T. delpontei* (5) (Table 3).

Four of the candidate models evaluated had ΔAICc ≤ 3 and showed the same ability to describe the data (Table 4). The best-ranked models included the effects of landscape disturbance, landscape environment (measured at 500 m in diameter), spatial arrangement of houses (density and distance) and refuge availability (number of peridomestic structures and number of chickens).

The average values of the model coefficients (Fig 3A) show that the chances of occurrence of triatomines associated with birds are 4 times higher in houses located in the disturbed landscape (OR = 4.43 CI 95% [0.94; 21.11]) and 3 times higher in houses located in the preserved landscape (OR = 3.29 CI 95% [0.83; 13.07]), compared with houses located in the intermediate disturbance landscape. In relation to the landscape metrics, the chances of invasion increase on average 4 times for each increase in hectare of forest cover (OR = 4.06 CI 95% [1.25; 12.93]) and up to 5 times for each increase in one unit of the DIVISION index (OR = 5.75 CI 95% [0.94; 35.16] in an area of influence of 500 m around the house. On the other hand, the chances of invasion increase on average 1.50 units for each increase in the number of peridomestic structures (OR = 1.50 95% CI [0.97; 2.31]) and number of chickens (OR = 1.49 95% CI [0.95; 2.36]). In contrast, the density of houses has a negative effect (OR = 0.60 95% CI [0.36; 1]) on the invasion odds (see Fig 4). The remaining predictors analyzed did not show consistent effects on the occurrence of invasion. For all the non-categorical predictors selected in these

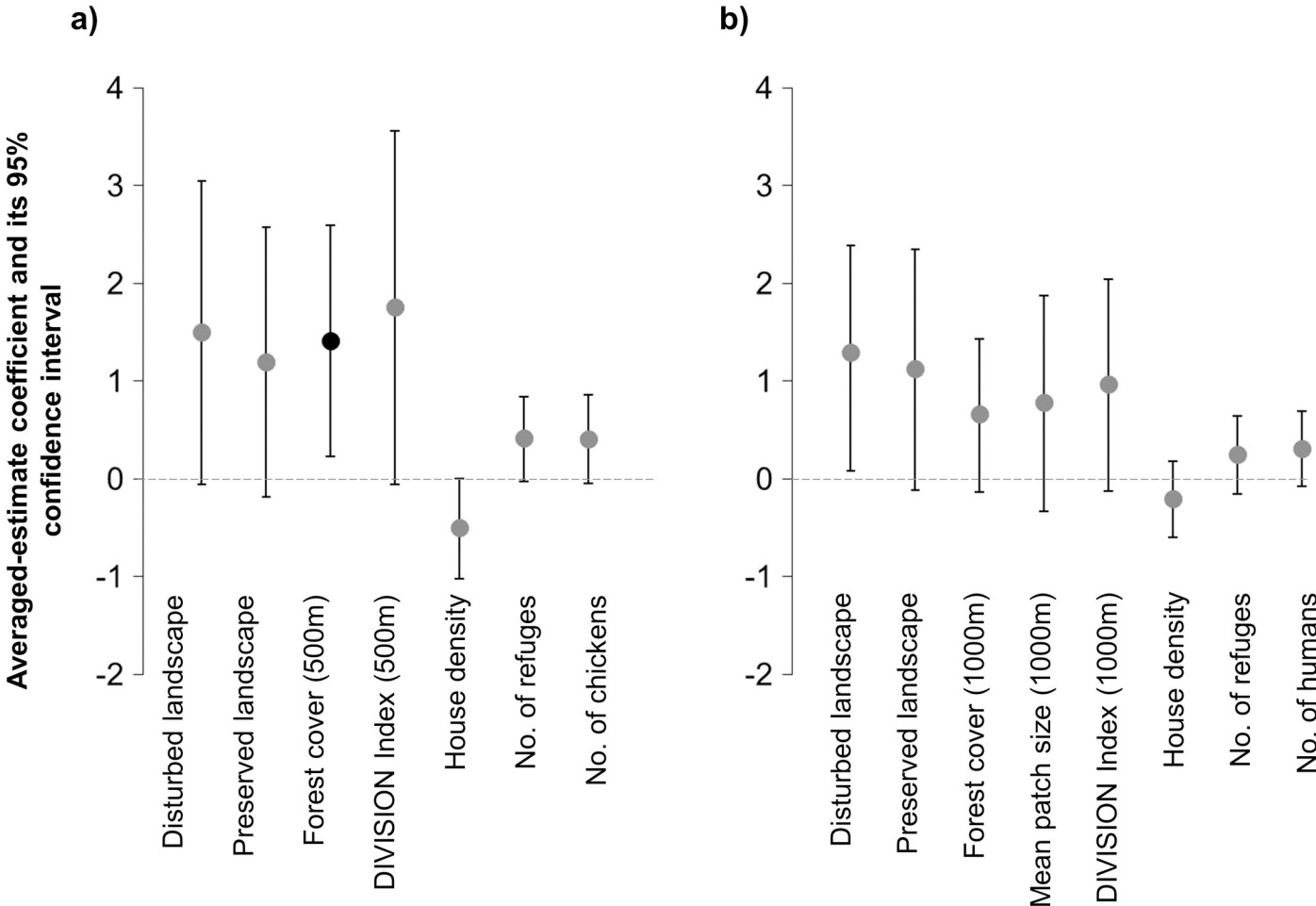

**Fig 3. Model-averaged estimates and confidence interval (CI 95%) of predictors effects on the invasion occurrence of triatomines associated with birds (A) and mammals (B).** Coefficient estimates (circles) are on log-odds scale. The black points correspond to the non-categorical predictors whose effects are different from 0 (95% confidence intervals do not contain the zero value). Categorical predictors coefficients must be interpreted as odd ratios considering the intermediate landscape as the reference level. The white point corresponds to categorical predictors whose effects are different from the reference level (95% CI do not contain OR = 1).

models, the 95% CI of the estimated coefficients included zero with the exception of the forest cover at 500m and housing density. In relation to the categorical predictors, the preserved and disturbed landscape includes 1 in their 95% CI, indicating that there would be no different effects between these landscapes on the occurrence of invasion.

The prediction errors estimated by cross validation are close to zero (Table 5). The model with the best predictive capacity (m49) is the one that contains the effects of environmental disturbance, landscape environment around the house and the spatial house arrangement as predictors (k = 10, prediction error = 0.43). According to the confusion matrix, the model correctly predicted 65.64% (86/131) of the observations, with a sensitivity of 0.49 (26/53) and a specificity of 0.76 (60/78), indicating that the classification errors correspond mostly to false negatives (type II error).

### Invasion occurrence: Mammalian-associated species

Among the evaluated houses, 60.3% (79/131) showed the presence of triatomines associated with mammals invading the domestic environment. The species most frequently recorded

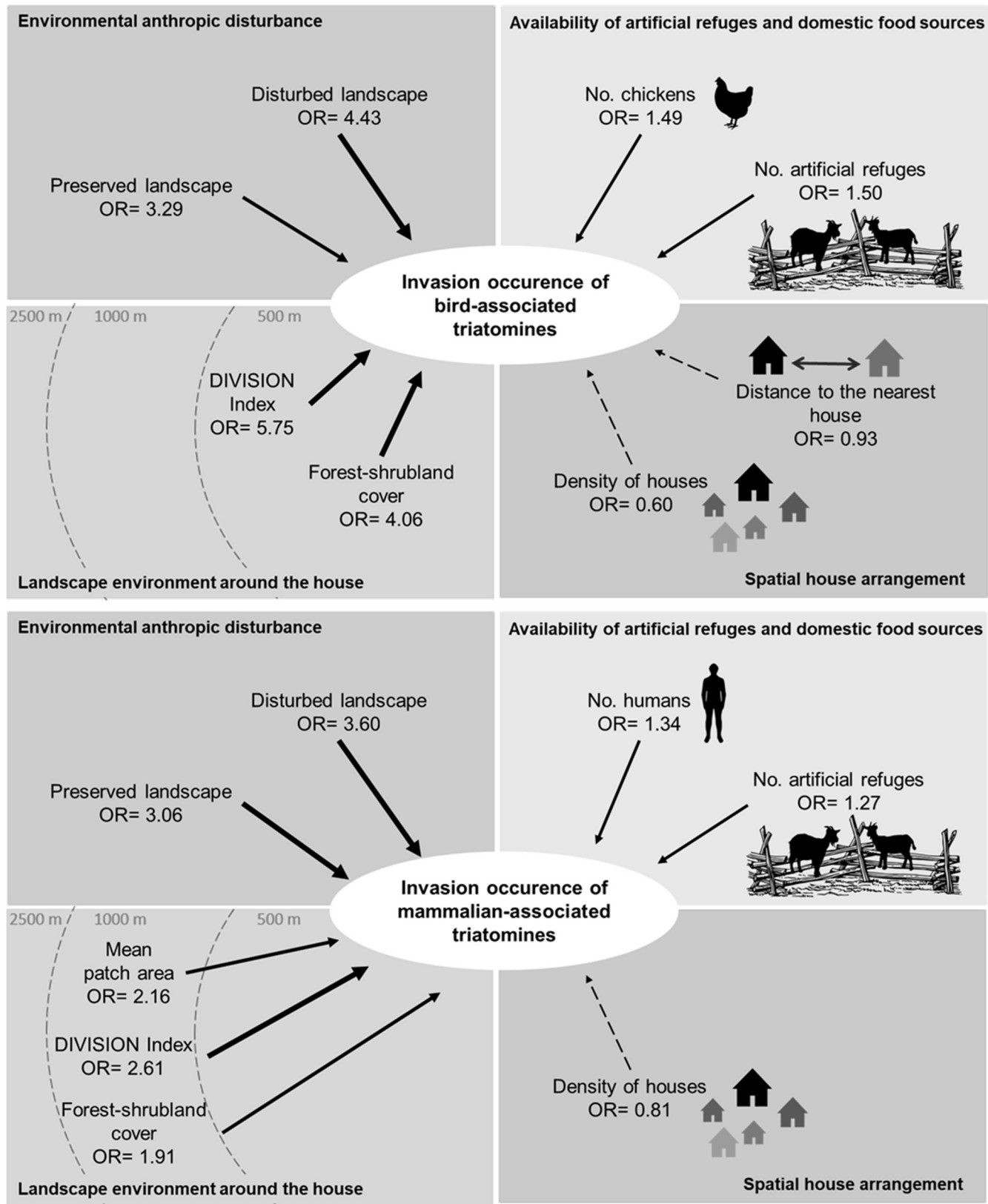

**Fig 4. Scheme of the four effects analyzed in the modeling and the main factors that influence the invasion occurrence of triatomines associated with birds and mammals in rural houses of the Chaco region of Córdoba.** Solid lines indicate positive effects, dashed lines indicate negative effects, and the

thickness of the arrows indicates the power of the observed effect on the response variable. OR (value of the average coefficient for each predictor on the Odds Ratios scale).

invading houses were *T. guasayana* (367), followed by *Panstrongylus guentheri* (23) and occasionally *T. breyeri* (3) (Table 3).

Of the sixty-nine candidate models evaluated, eight of them had ΔAICc ≤3.0 (Table 4). These best-ranked models included the effects of landscape disturbance, landscape environment (measure at 1000 m in diameter), spatial arrangement of houses (density) and refuge availability (number of peridomestic structures and number of humans) (Table 4).

The average values of the model coefficients (Fig 3B) show that the chances of occurrence of triatomine invasion is 3.60 times higher in houses located within a disturbed landscape (OR = 3.30 CI 95% [1.08; 10.91]) and 3.06 times higher in houses located within a preserved environment (OR = 3.06 CI 95% [0.89, 10.49]), compared with houses located within an intermediate disturbance landscape. Regarding landscape metrics, the chances of invasion increase on average 1.91 times for each increase in hectares of forest cover (OR = 1.91 CI 95% [0.87; 4.17]), up to 2.61 times for each increase in the unit of DIVISION index (OR = 2.61 CI 95% [0.88, 7.69]) and 2 times for each increase in hectare of the mean size of forest patch (OR = 2.16 CI 95% [0.71, 6.55]) in an area of influence of 1000 m around the house. Furthermore, the chances of invasion increase on average 1.30 units for each increase in the number of refuges available in peridomicile (OR = 1.27 95% CI [0.86, 1.89]) and number of inhabitants in the house (OR = 1.34 95% CI [0.92, 1.99]). In contrast, the density of houses has a negative effect on the invasion odds (OR = 0.81 CI 95% [0.55, 1.19]) (Fig 4). None of the best-ranked models included the landscape metrics measured at other scales or the other domestic animals evaluated. The remaining predictors did not show consistent effects on the occurrence of invasion. The prediction error estimated by cross validation are close to zero (Table 5). The model with the best predictive capacity (m46) includes the predictors of landscape disturbance and housing density (k = 10, prediction error = 0.36) According to the confusion matrix this model correctly predicts 64.88% of the observations (85/131), with a sensitivity of 0.87 (69/79) and a specificity of 0.30 (16/52), indicating that the classification errors correspond mostly to false positives (type I error).

**Table 5. Prediction error rate values obtained by K-fold cross validation and statistics of the confusion matrix calculated for the selected models of triatomine invasion occurrence in rural houses.**

| Response variable | Model | Prediction error (K = 10) | Global accuracy (%) | Sensitivity | Specificity |
|---|---|---|---|---|---|
| Invasion occurrence of bird-associated species | m41 | 0.39 | 64.88 | 0.35 | 0.84 |
| | m57 | 0.40 | 62.59 | 0.35 | 0.87 |
| | m38 | 0.42 | 60.30 | 0.30 | 0.87 |
| | m49 | 0.43 | 65.64 | 0.49 | 0.76 |
| Invasion occurrence of mammal-associated species | m49 | 0.38 | 64.12 | 0.91 | 0.22 |
| | m62 | 0.39 | 64.88 | 0.91 | 0.25 |
| | m4 | 0.36 | 64.12 | 0.89 | 0.25 |
| | m30 | 0.37 | 67.17 | 0.92 | 0.28 |
| | m6 | 0.40 | 60.70 | 1.0 | 0.0 |
| | m46 | 0.36 | 64.88 | 0.87 | 0.30 |
| | m14 | 0.40 | 60.70 | 1.0 | 0.0 |
| | m43 | 0.41 | 62.60 | 0.91 | 0.19 |

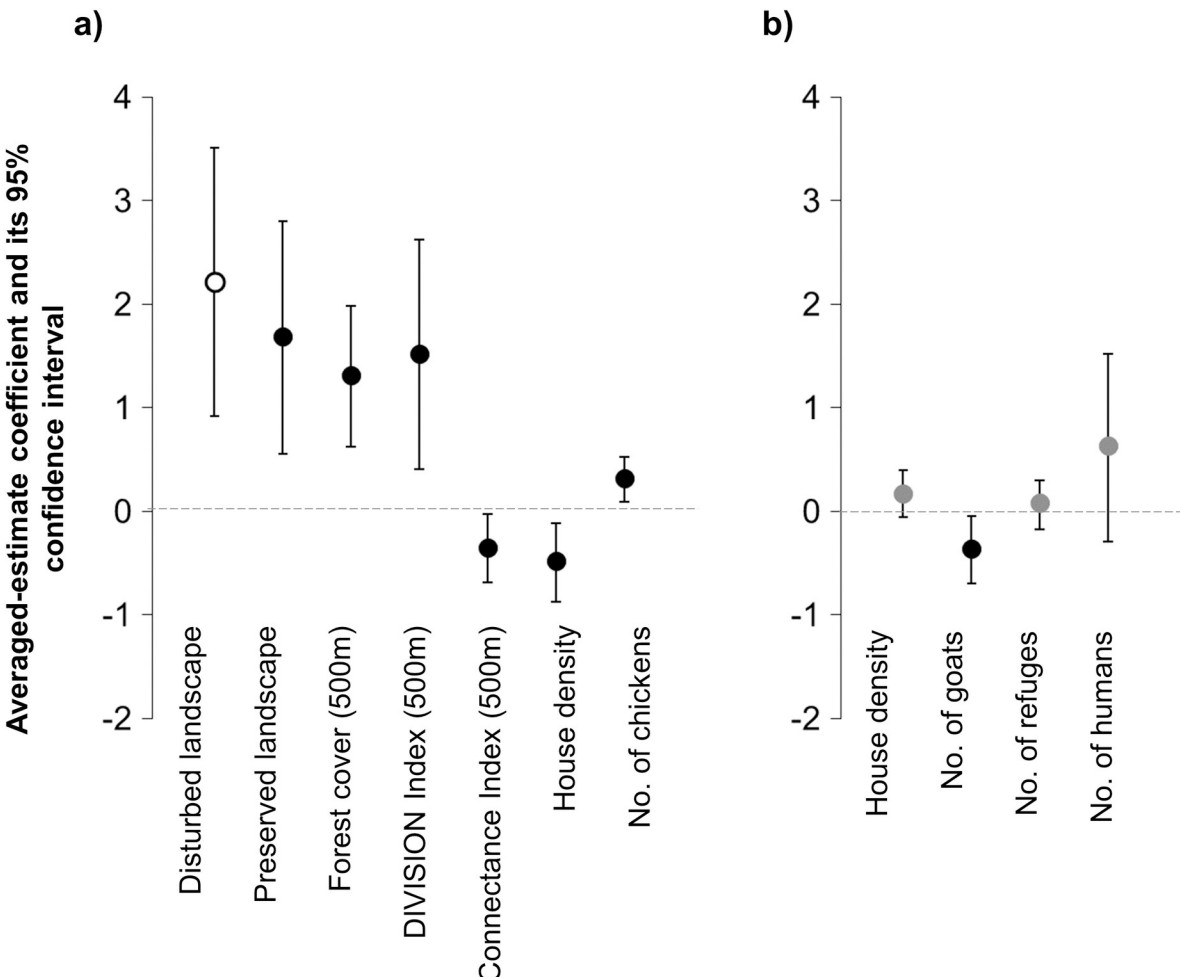

**Fig 5. Factor effect (log odd scale of standardized variables) over the frequency of triatomine invasion associated with birds (A) and mammals (B), estimated as weighted averages of the coefficients of models with ΔAICc<3. Vertical bars are 95% confidence intervals.** The black points correspond to the non-categorical predictors whose effects are different from 0 (95% confidence intervals do not contain the zero value). Categorical predictors coefficients must be interpreted as odd ratios considering the intermediate landscape as the reference level. The white point corresponds to categorical predictors whose effects are different from the reference level (95% CI do not contain OR = 1).

### Invasion frequency: Bird-associated species

Among the evaluated houses, 18.3% (24/131) showed invasion frequencies higher than 50% (invasion was reported at least half of the times the house was visited), a 19.1% of houses (25/131) presented invasion frequencies between 25–50% and a 60% of houses (78/131) did not register invasion by bird-associated triatomine species.

From the candidate models evaluated, only one fit the data adequately. The best-ranked model included the effects of landscape disturbance, landscape environment (measured at 500 m in diameter) and spatial arrangement of houses (density) (see Table 4).

The results show that the frequency of invasion is 9 times higher in houses located within a disturbed landscape (OR = 9.11 CI 95% [2.51; 33.45]) and almost 6 times higher in houses located within a preserved landscape (OR = 5.37 CI 95% [1.75; 16.45]), compared with houses located within an intermediate disturbance landscape. The frequency of invasion also increases on average almost 4 times for each increase in hectares of forest cover (OR = 3.67 CI 95%

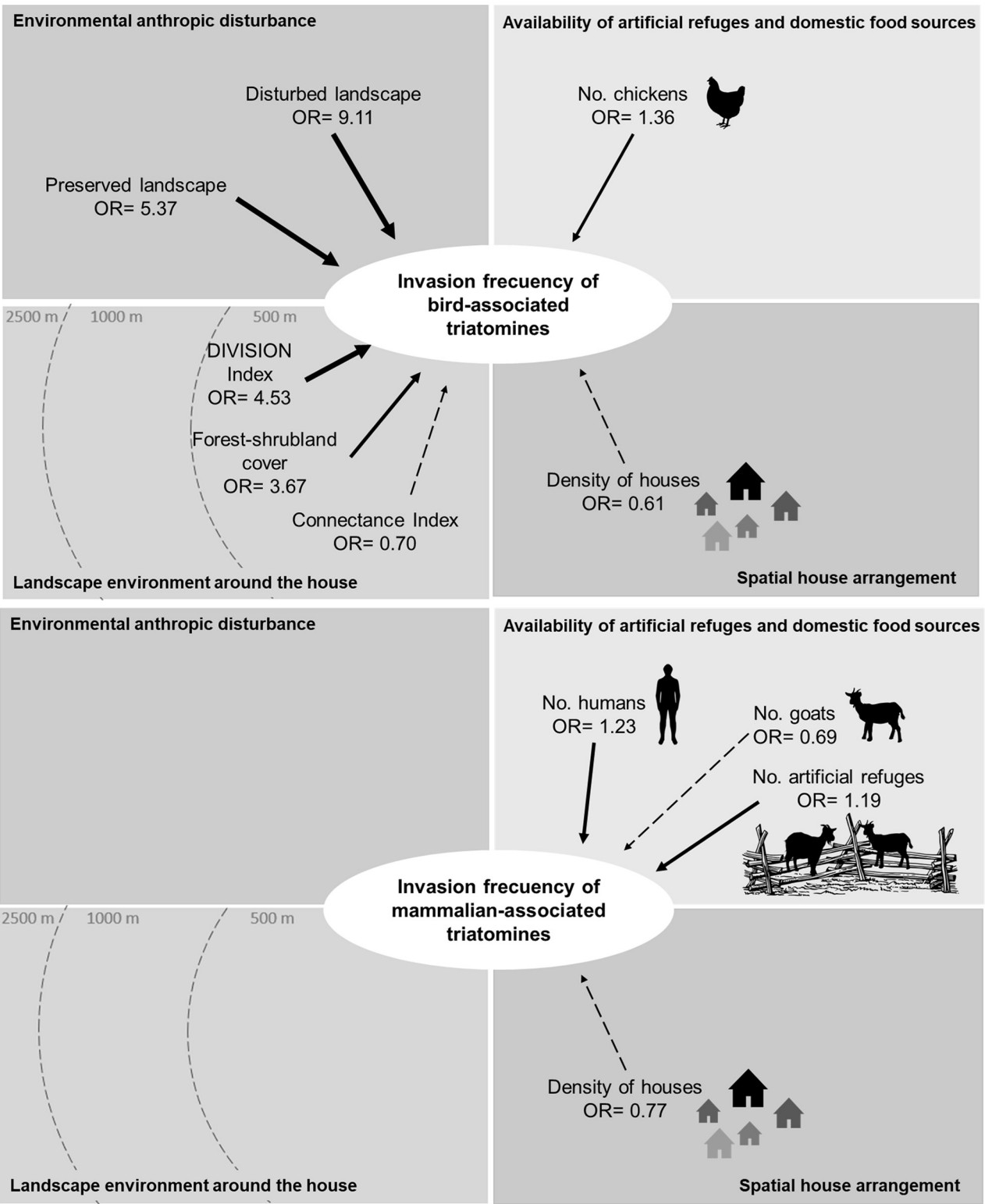

**Fig 6. Scheme of the four effects analyzed in the modeling and the main factors that influence the invasion frequency of triatomines associated with birds and mammals in rural houses of the Chaco region of Córdoba.** Solid lines indicate positive effects, dashed lines indicate negative effects, and the thickness of the arrows indicates the power of the observed effect on the response variable. OR (value of the average coefficient for each predictor on the Odds Ratios scale).

**Table 6. AUC-ROC values and statistics of the confusion matrix calculated for the selected models of triatomine invasion frequency in rural house.**

| Response variable | Model | AUC-ROC | Global accuracy (%) | Sensitivity | Specificity |
|---|---|---|---|---|---|
| Invasion frequency of bird-associated species | m39 | 0.71 | 0.67 | 0.36 | 0.89 |
| Invasion frequency of mammalian-associated species | m58 | 0.82 | 0.53 | 0.23 | 0.85 |
| | m34 | 0.75 | 0.48 | 0.00 | 1.0 |
| | m30 | 0.64 | 0.50 | 0.11 | 0.90 |

[1.86; 7.24]) and 5 times for each increase in one unit of the DIVISION index (OR = 4.53 95% CI [1.51; 13.74]. Connectivity between patches of forest cover at 500 m (OR = 0.70 95% CI [0.50; 0.97]) and the density of houses (OR = 0.61 95% CI [0.42; 0.89]) show a negative effect on the frequency of triatomine invasion (Fig 5A). Although the selected model did not include predictors related to the availability of refuges or domestic meals, the average estimated coefficient for the number of chickens in the peridomicile area also shows a positive effect on the frequency of invasion (OR 1.36 CI 95% [1.09; 1.70]) (Fig 6).

The validation of the selected model had an AUC value of 0.71, indicating that the selected model has good predictive capacity (Table 6). According to the confusion matrix, this model correctly predicts 67% of the observations (87/129), with a sensitivity of 0.36 (19/53) and a specificity of 0.89 (68/76), indicating that the classification errors correspond mainly to false negatives (type II errors).

### Invasion frequency: Mammalian-associated species

A total of 25.6% (33/129) of the houses showed invasion frequencies higher than 50%, 26.3% (34/129) of houses showed invasion frequencies between 25–50% and 38.7% (50/129) did not report invasion on any occasion by mammalian-associated triatomine species.

Three of the candidate models evaluated were selected according to the Akaike's criteria (ΔAIC <2.0). The best-ranked models included the effects of refuge availability and food sources (number of goats, number of humans and number of peridomestic structures) and spatial arrangement of houses (density) (Table 4). None of these models included predictors linked to landscape metrics or landscape disturbance.

The results show that the frequency of invasion increases on average 1.2 times for each increase in the number of peridomestic structures (OR = 1.19 CI 95% [0.94; 1.49]) and in the number of humans (OR = 1.23 95% CI [0.97; 1.57]). The density of houses shows a negative effect on the frequency of invasion (OR = 0.77 CI 95% [0.57; 1.04] together with the number of goats present in the peridomicile (OR = 0.69 CI 95% [0.50; 0.95]) (Figs 5B and 6).

The validation of the model (m58) that includes the predictors of density, number of goats and number of humans had a value of AUC = 0.82, indicating a good predictive capacity. The remaining two models (m30 and m34) showed a comparatively lower discriminatory capacity between classes (AUC = 0.75 and AUC = 0.64 respectively) (Table 6). Based on the confusion matrix, the model 58 correctly predicted 53% (69/129) of the observations with a sensitivity of 0.23 (16/67) and a specificity of 0.85 (53/62). In general, the selected models showed high specificity, but low sensitivity, indicating that the classification errors correspond mainly to false negatives (type II errors).

### Discussion

The study documents the occurrence of seven species of triatomines–*T. infestans*, *T. guasayana*, *T. garciabesi*, *T. platensis*, *T. delpontei*, *T. breyeri* and *P. guentheri*—actively dispersing into rural houses in the northwestern Chaco region of Córdoba Province. The sylvatic species most frequently collected were *T. guasayana* and *T. garciabesi* (Table 3).

Previous studies carried out in the same region, during 1962 and 1984, had reported three sylvatic species -*T. breyeri*, *T. guasayana* and *T. eratirusyformis*-, invading houses but without colonizing them [56]. A more recent study in the same area also reported the presence of *T. guasayana* and *T. garciabesi* in peridomicile structures [57].

Based on our field surveys conducted between 2017–2020, the sylvatic triatomine species associated with mammals (*T. guasayana*, *T. breyeri* and *P. guentheri*) were recorded invading a higher number of rural houses in the region and with higher frequency than the species associated with birds (*T. garciabesi*, *T. platensis* and *T. delpontei*).

The evaluation of the environmental anthropic disturbance hypothesis, shows a positive effect of the landscape disturbance on the invasion of sylvatic triatomines. This effect proved to be stronger on the invasion occurrence and frequency of species associated with birds than in the occurrence of species associated with mammals (Figs 3 and 5). However, we found no evidence of positive association between landscape disturbance and the invasion frequency of triatomines associated with mammals (Fig 6). According to the analyses, the fragmentation of the forest habitat in the Chaco region would favor the active dispersal of triatomines increasing the chances of domicile invasion. Our results suggest that even under the suboptimal conditions (loss of wild hosts and potential refuges), heavily disturbed environments support higher indices of triatomines dispersal than it was hypothesized. Coinciding with previous studies [48,58] we found that anthropic disturbances in the habitat are associated with increased abundance of vectors, as is the case of *Rhodnius pallescens* in Panama. However, similar studies carried out in Brazil suggest that the dispersal of some sylvatic species, such as *Rhodnius neglectus* and *Panstrongylus geniculatus* in the Cerrado region, or *Triatoma tibiamaculata* in the Atlantic Forest region, occurs more frequently in areas of intermediate disturbance [49,59]. On the other hand, Rocha-Leite et al. [60] suggest that the less degraded areas maintain larger populations of triatomines and, therefore, present higher rates of dispersal and house invasion.

In addition, the models showed that the occurrence of triatomines invasion and the frequency of invasion (in the case of triatomines associated with birds) would be mostly associated with the effect of habitat quality (coverage) and degree of isolation (subdivision) of the fragments of forest-shrub cover surrounding the house. Hence, this may suggest that patches of fragmented areas, which are in proximity to the domestic environment, could function as sources of sylvatic populations of triatomines favoring the invasion. One possible explanation is that dispersal rates are higher in fragmented landscapes, because hungry triatomines show a higher dispersal potential than fed insects, increasing the chances of invasion of dwellings in search of food [50,58].

The effect of landscape metrics, for triatomines associated with birds, is consistent only in the area of influence of a circle with 500 meters in diameter. In that area, the average size of forest patches is between 1–2 hectares. According to previous studies, patches between 1–2 hectares can support assemblages of common birds, like specimens of Furnariidae and Psittacidae, whose nests are used as wild refuges by triatomines [26,38,61]. For the group of triatomines associated with mammals, the landscape effect is consistent in the area of influence of 1000 meters in diameter. This area contains on average larger forest patches, between 1 and 10 hectares. So, the observed effect could be associated with the fact that mammal species linked to the wild transmission cycle of *T. cruzi* have requirements for habitat size ranging from 4 hectares (*Didelphis albiventris* and *Chaetophractus spp.*) up to 200 hectares (*Conepatus chinga*), besides, their refuges are not established as close to houses as bird nests.

The results also evidenced an association between house invasion and the availability of refuges and food sources in the peridomicile area. The number of chickens is positively associated with the invasion of bird-associated triatomines. Previous studies reported that dispersing individuals of *T. garciabesi* and *T. platensis* show preference for peridomestic ecotopes

associated with birds (hen houses and trunks of the genus *Prosopis spp.* where chickens sleep) [39,62]. On the other hand, the invasion of triatomines associated with mammals showed a positive association with the number of inhabitants in the house. This effect is consistent with previous studies carried out in the north of Argentina, where the food sources of *T. guasayana* specimens captured in the wild were analyzed, observing that they fed in a higher proportion on domestic animals than wild animals, showing a high rate feeding on humans [16,40].

Contrary to expectations, the number of goats in the peridomicile was associated with a lower frequency of house invasion. However, this result must be interpreted taking into account that 70% (91/129) of the sampled households do not have goats in the peridomicile area and only a minority are producers. Furthermore, previous studies indicated that the distribution of triatomines associated with mammals, such as *T. guasayana*, is strongly associated with the density of goats [63]. Hence, the effect of this variable should be interpreted keeping in mind the limitations related to the few records in the area, variations in the number of specimens, between samplings, due to productive activity, the distance at which the corral is located from the house and their construction characteristics, among other factors that could be intervening.

Finally, we examined the hypothesis that a higher density of houses favors the spatial concentration of physical and chemical signals (light from dwellings and domestic hosts odors) that would attract dispersant triatomines. We observed that the active dispersal of sylvatic triatomines occurs mainly in houses isolated from each other. In the study area, the highest house density values correspond to "neighborhoods", built in small building lots, where families rarely build peridomestic structures for animals, therefore the chemical clustering signals do not occur. In relation to physical signals, other studies pointed out that the occurrence of more dispersed dwellings within the landscape favors the perception of artificial light by dispersing insects, also observing that the flight initiation of sylvatic specimens, such as *Rhodnius prolixus*, is independent of the number of artificial lights available in the environment [60,64].

In summary, the active dispersal of sylvatic triatomines towards the domestic environment can be divided into two stages, the first has to do with the beginning of flight activity, which responds to physiological modulators that trigger the search for food, shelter and reproduction [29,65]. The second stage has to do with the orientation of the flight and the approach of triatomines to the anthropic habitat [60].

Our results show that in the Chaco region of Córdoba, invasion of sylvatic triatomines apparently occurs with higher frequency in disturbed landscapes, with houses spatially isolated and in proximity to subdivided fragments of forest-shrub cover. The availability of domestic refuges in the peridomicile structures as well as the presence in a great number of domestic animals and humans enhance the chances of triatomines invasion.

Since these species are not subject to chemical control by the control vector programs, their approach to the domestic environment represents a potential epidemiological risk. Therefore, the understanding and study of the ecological, physiological and behavioral factors that modulate the active dispersal of sylvatic triatomines to the domestic environment deserves more attention.

The results obtained about the effect of environmental anthropic disturbance must be carefully interpreted due to limitations in the sample size of the data obtained. The sample size of the houses grouped in landscapes with intermediate disturbance (n = 21) is smaller than the sample size of the houses grouped in disturbed (n = 68) and preserved environments (n = 42) (Table 2). Therefore, we cannot demonstrate ample differences in triatomines invasion between the environments due to the wide 95% CI estimated. However, we can interpret based on the size effects estimated that landscape disturbance has a relevant influence on the invasion phenomena [66]. In general, the internal validation of the models showed a good predictive

capacity, with global accuracy greater than 60% and AUC values greater than 0.70. Nevertheless, the interpretation of the models should be made with caution, since they offer an approximate explanation of the effect of some ecological factors on the complex phenomenon of active dispersal of sylvatic triatomines.

## Supporting information

**S1 Data. Raw data on invasion occurrence of bird-associated and mammals-associated triatomines are provided in the sheets named "Bird-associated spp occurrence" and "Mammal-associated spp occurrence", respectively.** Raw data on invasion frequency of bird-associated and mammals-associated triatomines are provided in the sheets named "Bird-associated spp frequency" and "Mammal-associated spp frequency", respectively. The file includes a "Variables" sheet with the description of each variable.
(XLSX)

**S1 Fig. Logistic regression model of invasion occurrence of bird-associated triatomines and factors with the highest effect.** The blue line corresponds to the predicted values, the gray band its 95% confidence intervals, the gray points are the partial residuals, and the upper and lower lines are the observed values of the response variable.
(TIF)

**S2 Fig. Logistic regression model of invasion occurrence of mammals-associated triatomines and factors with the highest effect.** The blue line corresponds to the predicted values, the gray band its 95% confidence intervals, the gray points are the partial residuals, and the upper and lower lines are the observed values of the response variable.
(TIF)

**S3 Fig. Logistic regression model of invasion frequency of bird-associated triatomines and the factors with the highest effect.** The blue line corresponds to the predicted values, the gray band its 95% confidence intervals, the gray points are the partial residuals, and the upper and lower lines are the observed values of the response variable.
(TIF)

**S4 Fig. Logistic regression model of invasion frequency of mammals-associated triatomines and the factors with the highest effect.** The blue line corresponds to the predicted values, the gray band its 95% confidence intervals, the gray points are the partial residuals, and the upper and lower lines are the observed values of the response variable.
(TIF)

## Acknowledgments

We thank the Provincial Chagas Program of Córdoba (PPCh) and the members of the Movimiento Campesino de Córdoba (MoCC) for the valuable information provided for the selection of the study area and for their collaboration during field samplings. We also thank Claudia Rodriguez, Fernando Carezzano, Ana López and Emilia Secaccini for their collaboration in field and laboratory activities. We specially acknowledge the local communities for their support and participation.

## Author Contributions

**Conceptualization:** Miriam Cardozo, Liliana Beatríz Crocco, David Eladio Gorla.

**Data curation:** Miriam Cardozo, David Eladio Gorla.

**Formal analysis:** Miriam Cardozo, David Eladio Gorla.

**Funding acquisition:** David Eladio Gorla.

**Investigation:** Miriam Cardozo, Federico Gastón Fiad, Liliana Beatríz Crocco, David Eladio Gorla.

**Methodology:** Miriam Cardozo, Federico Gastón Fiad, Liliana Beatríz Crocco, David Eladio Gorla.

**Project administration:** Liliana Beatríz Crocco, David Eladio Gorla.

**Resources:** Liliana Beatríz Crocco.

**Supervision:** Liliana Beatríz Crocco, David Eladio Gorla.

**Validation:** Miriam Cardozo, David Eladio Gorla.

**Visualization:** Miriam Cardozo, Liliana Beatríz Crocco, David Eladio Gorla.

**Writing – original draft:** Miriam Cardozo.

**Writing – review & editing:** Miriam Cardozo, Federico Gastón Fiad, Liliana Beatríz Crocco, David Eladio Gorla.

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
