## [Decision Letter · Decision Letter 0]

26 Apr 2021

Dear Biol. Cardozo,

Thank you very much for submitting your manuscript "Modeling the effect of habitat fragmentation on rural house invasion by sylvatic triatomines: a multiple landscape-scale approach" for consideration at PLOS Neglected Tropical Diseases. As with all papers reviewed by the journal, your manuscript was reviewed by members of the editorial board and by several independent reviewers. In light of the reviews (below this email), we would like to invite the resubmission of a significantly-revised version that takes into account the reviewers' comments. 

The Second reviewer has expressed major reservations about the thematic aspects your MS. Please review your MS to establish if the MS indeed incorporates cardinal modeling concepts and adjust accordingly.

We cannot make any decision about publication until we have seen the revised manuscript and your response to the reviewers' comments. Your revised manuscript is also likely to be sent to reviewers for further evaluation.

Sincerely,

Paul O. Mireji, PhD

Associate Editor

Bruce Lee

Deputy Editor

The Second reviewer has expressed major reservations about the thematic aspects your MS. Please review your MS to establish if the MS indeed incorporates cardinal modeling concepts and adjust accordingly.

Reviewer's Responses to Questions

**Key Review Criteria Required for Acceptance?**

**Methods**

-Are the objectives of the study clearly articulated with a clear testable hypothesis stated?

-Is the study design appropriate to address the stated objectives?

-Is the population clearly described and appropriate for the hypothesis being tested?

-Is the sample size sufficient to ensure adequate power to address the hypothesis being tested?

-Were correct statistical analysis used to support conclusions?

-Are there concerns about ethical or regulatory requirements being met?

Reviewer #1: The study was built on local observations on increased numbers of sylvatic triatomines in domestic human environments in the relatively dry western Chaco Region of Argentina. The objectives of the study were built on the hypothesis that gross anthropic changes of the ecosystem, particularly the fragmentation of the habitat, major alterations in the availability of food sources and refuges, and increased number of domestic animals, is a major cause of the increase in domicile shift of the vectors of Trypanosoma cruzi. In my view, the overall design of the study adequately addressed the stated objectives, the sample sizes were sufficient, and appropriate statistical analyses of the data generated were used. The house inhabitants who participated in the study gave oral consent, and were trained in the identification and collection of triatomines, and avoidance of infection by T. cruzi.

Reviewer #2: The title of the manuscript is not fully reflecting it contain. The authors indicated that the manuscript report a modeling experiments, however the text did not put emphasize of modelling studies either in the introduction or in the discussion. No comparison with existing models was conducted. I will suggest to authors to replace the word “modeling” in the title by “Enhancing the understanding”. 

The methodology is about study sites, and data analysis and not modelling experiments and the whole line 547-552 should be remove from the text. The text further lack justification why the selected methods were used for the analysis. 

Why the maximum likelihood method for classification was used instead of more robust technique such as random forest? Please justify the choice of the methodology 

Although English is good, the use of tenses should be handling with caution.

Reviewer #3: --Are the objectives of the study clearly articulated with a clear testable hypothesis stated? yes

Are the objectives of the study clearly articulated with a clear test- yes

-Is the population clearly described and appropriate for the hypothesis being tested? yes

-Is the sample size sufficient to ensure adequate power to address the hypothesis being tested? yes for the most part

-Were correct statistical analysis used to support conclusions? yes- adequate

-Are there concerns about ethical or regulatory requirements being met?- no IRB at the universities so did not have to go through IRB- everything seems fine

**Results**

-Does the analysis presented match the analysis plan?

-Are the results clearly and completely presented?

-Are the figures (Tables, Images) of sufficient quality for clarity?

Reviewer #1: The results presented are interesting and adequately match the analyses plans. Results are also clear and well presented in different Tables, and adequately discussed. Perhaps Fig 2 may need to be improved for better visual clarity.

Reviewer #2: The title of the manuscript is "Modeling the effect of habitat fragmentation on rural house invasion by sylvatics triatomines: a multiple landscape-scale approach"however the text describes the occurrence and frequency of sylvatic triatomines in rural houses, and evaluates the effect of habitat fragmentation and other ecological factors in regards to the phenomena. A number of analysis were used to explain the mechanism resulting to the species invasion; however, each method revealed some level of limitation and the authors suggested an integrated approach which can better capture of the complexity of the system at hands and provide more insight. 

The Tables and Figures are of good qualities

I will suggest change on the title and rephrasing certain sections of the text

Reviewer #3: -Does the analysis presented match the analysis plan? yes

-Are the results clearly and completely presented? yes

-Are the figures (Tables, Images) of sufficient quality for clarity?-- some of the figures (graphs) appear a bit pixellated- but that could be the quality of my computer- would require prof. editorial check at PlosNTDS

I mention some modifications to table 2 suggested (see comments below) in summary and general comments.

**Conclusions**

-Are the conclusions supported by the data presented?

-Are the limitations of analysis clearly described?

-Do the authors discuss how these data can be helpful to advance our understanding of the topic under study?

-Is public health relevance addressed?

Reviewer #1: The conclusions are adequately supported by the data presented and scope of analyses are well described. The Discussion captures the background that led to the work and the implication of the results, and specifically their downstream implication as well as the gross significance of anthropic changes in the environment.

Reviewer #2: The conclusions are well align with the majority of the text, however the issue of modelling raised above should be addressed

Reviewer #3: -Are the conclusions supported by the data presented?

-Are the limitations of analysis clearly described? yes but really the T. cruzi results tested microscopically add nothing to this study so I suggest that they don't include this (only microscopic analysis was done on some bugs and actually molecular analysis is at this point the 'norm' as there are too many false negatives with microscopic analysis) 

-Do the authors discuss how these data can be helpful to advance our understanding of the topic under study? yes

-Is public health relevance addressed? yes-the household construction could be described in more detail (see below in my general comments to authors)

**Editorial and Data Presentation Modifications?**

Reviewer #1: Apart from the need to improve Fig 2, I recommend that 'Accept the manuscript as it is'.

Reviewer #2: Major revision especially on the contain focusing on the new title I proposed or something similar

Reviewer #3: See below the editorial modification in summary and general comments)

**Summary and General Comments**

Reviewer #1: A good study well carried out.

Reviewer #2: This is a nice study that need some fine tuning before getting published

Reviewer #3: This is a very interesting paper that evaluates anthropogenic landscape structure and configuration (this evaluation of spatial configuration of habitat as well as houses is quite novel and I applaud the authors for this ) on triatomine home invasions. I particularly like analyses being parsed out for triatomine species that are predominantly avian associated and those that are mammal associated. Scientists that study triatomines will find a lot of value and food for thought and discussion with the analyses done in this paper. 

However, there are some issues that should be addressed. 

There are some grammar errors in english (e.g. lots of subject-verb agreement errors)- I am sure I only caught a few and not all of these errors so this should be double chcked by English editors. 

line 168- only looking for T. cruzi by microscopic evaluation of bug feces can markedly underestimate infection rates , multiple studies confirm this. you may want to not present this microscopic data without molecular confirmation. 

For table 2- can you please show the totals across each landscape (adding up the communities for each habitat type?)- it would be helpful to know the mean number of goats, chickens, dogs, etc for each household not just the total across all households.

Do all households have electricity? 

In the methods, can you describe a bit more what a 'typical' house might look like (construction, lights, windows, permeability to invasion of bugs?)

Minor comments

line 131- change 'refuges' to 'refuge'

line 128- 'increases' change to 'increase' 

line 146- can you descrine the patterns of habitat loss in Cordobal province?

line 164- why were these seasons chosen for field sampling?

line 591- put a comma after 'besides'

PLOS authors have the option to publish the peer review history of their article (what does this mean?). If published, this will include your full peer review and any attached files.

Reviewer #1: Yes: Ahmed Hassanali

Reviewer #2: Yes: HENRI EDOUARD ZEFACK TONNANG

Reviewer #3: No
---

## [Decision Letter · Decision Letter 1]

21 Jun 2021

Dear Biol. Cardozo,

We are pleased to inform you that your manuscript 'Effect of habitat fragmentation on rural house invasion by sylvatic triatomines: a multiple landscape-scale approach' has been provisionally accepted for publication in PLOS Neglected Tropical Diseases.

Best regards,

Paul O. Mireji, PhD

Associate Editor

Bruce Lee

Deputy Editor

Reviewer's Responses to Questions

**Key Review Criteria Required for Acceptance?**

**Methods**

-Are the objectives of the study clearly articulated with a clear testable hypothesis stated?

-Is the study design appropriate to address the stated objectives?

-Is the population clearly described and appropriate for the hypothesis being tested?

-Is the sample size sufficient to ensure adequate power to address the hypothesis being tested?

-Were correct statistical analysis used to support conclusions?

-Are there concerns about ethical or regulatory requirements being met?

Reviewer #2: The methodology include study sites and data analysis techniques what is an improvement as it better corroborate with the new title of the manuscript. It will nice if the text could include justification why the selected methods were used for the analysis. The use of English tenses should be handling with caution.

Reviewer #3: Yes to all above questions.

**Results**

-Does the analysis presented match the analysis plan?

-Are the results clearly and completely presented?

-Are the figures (Tables, Images) of sufficient quality for clarity?

Reviewer #2: The results section is very well elaborated, the only challenge is the use of English tenses. Rare way of starting a manuscript with the word “after”

Reviewer #3: Yes to all above questions

**Conclusions**

-Are the conclusions supported by the data presented?

-Are the limitations of analysis clearly described?

-Do the authors discuss how these data can be helpful to advance our understanding of the topic under study?

-Is public health relevance addressed?

Reviewer #2: Yes the conclusion is well elaborated and align with the rest of the text. Proper analysis were conducted.

Reviewer #3: Yes to all above questions

**Editorial and Data Presentation Modifications?**

Reviewer #2: The key issue in the overall manuscript is the use of English languages tenses.

Reviewer #3: In the Abstract

You could omit the term 'with binomial error distribution and logit link function' but you were evaluating the effect of independent variables in i-iv on the invasion or presence of sylvatic triatomines in houses, correct? please state what you were evaluating the 'effect of' (independent variables) on the dependent variable-state what the dependent variable It seems to be missing in this sentence so after iv, put 'on triatomine invasion into homes'

Please omit 'For the first time we present evidence suggesting' and replace with 'Study data suggest that invasion with ....'

put an apostrophe after triatomines' in the last sentence.

Introduction- some minor comments.

line 86- after 'Argentina' omit the comma

line 88- change 'rate' to 'rates'

line 90 omit the comma after 'triatomines'

line 107- replace 'refuges' with 'refuge'

118- omit the comma after 'insects'

line 132- change -'effect' to 'effects'

**Summary and General Comments**

Reviewer #2: Overall the paper is reporting a relevant topic to the journal and the authors have conducted adequate analysis which in my view will bring additional insight to the readers.

Reviewer #3: I think most comments have been adequately addressed- there might be some minor grammatical errors that are still there that I didn't catch. I had very few additional comments of editorial detail nature.

PLOS authors have the option to publish the peer review history of their article (what does this mean?). If published, this will include your full peer review and any attached files.

Reviewer #2: No

Reviewer #3: No

---

## [Editor Report · Acceptance letter]

8 Jul 2021

Dear Biol. Cardozo,

We are delighted to inform you that your manuscript, "Effect of habitat fragmentation on rural house invasion by sylvatic triatomines: a multiple landscape-scale approach," has been formally accepted for publication in PLOS Neglected Tropical Diseases.

Best regards,

Shaden Kamhawi

co-Editor-in-Chief

Paul Brindley

co-Editor-in-Chief
